# A Novel Eulerian Model Based on Central Moments to Simulate Age and Reactivity Continua Interacting with Mixing Processes

Jurjen Rooze[1], Heewon Jung[2], and Hagen Radtke[1]

[1]Department of Physical Oceanography and Instrumentation, Leibniz Institute for Baltic Sea Research (IOW), Warnemünde, Germany.
[2]Department of Geological Sciences, Chungnam National University, Daejeon, South Korea.

**Correspondence:** Jurjen Rooze (jurjen.rooze@io-warnemuende.de)

**Abstract.** In geoscientific models, simulating the properties associated with particles in a continuum can serve many scientific purposes, and this has commonly been addressed using Lagrangian models. As an alternative approach, we present an Eulerian method here: Diffusion-advection-reaction type of partial differential equations are derived for centralized moments, which can describe the distribution of properties associated with chemicals in reaction-transport models. When the property is age, the equations for centralized moments (unlike non-central moments) do not require terms to account for aging, making this method suitable for modeling age tracers. The properties described by the distributions may also represent kinetic variables affecting reaction rates. In practical applications, continuous distributions of ages and reactivities are resolved to simulate organic matter mineralization in surficial sediments, where macrofaunal and physical mixing processes typically dominate transport. In test simulations, mixing emerged as the predominant factor shaping reactivity and age distributions. Furthermore, the applications showcase the method's aptitude for modeling continua in mixed environments while also highlighting practical considerations and challenges.

## 1 Introduction

The partial differential equation (PDE) to describe chemical diffusion (Fick, 1855) is mathematically equivalent to Fourier's heat conduction equation (Fourier et al., 1822), which has become ubiquitous in science to describe random transport processes (Narasimhan, 1999). When materials are transported, tracking other associated properties besides the concentration can be desirable. For example, the age (or transit time) of fluids and chemicals has often been simulated. Diffusive mixing will lead to a local spreading in ages. Resolving these distributions is commonly achieved through Lagrangian approaches, which aim to simulate sufficiently large numbers of individual particles to describe the evolution of the properties and their statistical distributions. Alternatively, Eulerian approaches utilizing PDEs offer the advantage of analytical evaluation and are computationally less expensive. Analytical solutions for age distributions in particular boundary condition problems can be found in Delhez and Deleersnijder (2002) and Kuderer (2022). Deleersnijder et al. (2001) and Delhez and Deleersnijder (2002) derived Eulerian PDEs to simulate the effect of diffusion on the mean and higher raw moments and considered the effect of radioactive decay on age distributions. In this study, we derive Eulerian PDEs for centralized moments. These are more readily intuitively understood than raw moments and are not affected by aging, making them ideal for modeling time tracers.

Beyond modeling passive tracers, we intend to test moment-based PDEs in more complex applications whereby chemical reactions depend on and affect distributions. Modeling the effect of "aging" on the apparent organic matter reactivity (Middelburg, 1989, 2019) provides an interesting practical case study. Bulk organic matter in sediments and soils contains materials with varying reactivities (e.g., De Leeuw and Largeau, 1993), and the bulk degradation rate depends on the entire matrix. As more reactive components disappear first, the remaining organic matter becomes more refractory (Zonneveld et al., 2010).

Organic molecules also undergo transformations, which generally lower the reactivity (Burdige, 2007). The overall decreasing reactivity over time is contained in the concept of aging. In multi-G models, separate state variables represent discrete classes of varying reactivities (Jørgensen, 1978; Westrich and Berner, 1984). The disadvantage of this approach is that reactivity classes and their distribution in deposited organic matter are somewhat arbitrarily chosen (Jørgensen, 1978), resulting in parameterizations that are difficult to compare between studies. Also, no more than three classes are usually defined, which cannot represent

more gradual changes in reactivity. The reactivity of organic matter may also be described as a continuum for which various distribution functions have been proposed (e.g., Boudreau and Ruddick, 1991; Vähätalo et al., 2010; Xu et al., 2022). The gamma distribution is most commonly used (Arndt et al., 2013; Freitas et al., 2021), in part because it allows an analytical solution for the evolution of the continuum over time (Boudreau and Ruddick, 1991). It can be easily implemented in sediment models by replacing time with sediment depth based on the assumption of a constant burial velocity or a reconstruction of

the deposition history. However, the space-for-time substitution only accounts for the burial of particulate organic matter and ignores mixing processes.

    Animals and plants continuously cause disturbances in sediments, which is referred to as bioturbation in literature (Meysman et al., 2006). Bioturbation typically dominates the transport of solids in sediments up to a depth of $\sim 10\,\mathrm{cm}$ (Tromp et al., 1995; Middelburg et al., 1997; Boudreau, 1994). In reaction-transport models, this process is most commonly implemented as Fickian

transport (Goldberg and Koide, 1962; Guinasso and Schink, 1975; Meysman et al., 2005), i.e., as chemical diffusion, but with a diffusivity decreasing over depth. Mixing of particulate organic matter can also be the result of other natural processes or anthropogenic activities, such as trawl fishing (e.g., De Borger et al., 2021). The inability of reactivity continuum models to account for mixing either caused an error or limited their application to environments without significant mixing processes (Freitas et al., 2021). Previously, Lagrangian methods have been developed to simulate organic matter mineralization in turbated

sediments (Meile and Van Cappellen, 2005; Kuderer, 2022), but these have only included reaction networks with few chemicals. The development of an alternative Eulerian approach, compatible with classical early diagenetic reaction-transport models (Wang and Van Cappellen, 1996; Boudreau, 1997), may be needed to encourage wider usage of modeled reactivity continua in turbated environments.

    Here, we first derive expressions that describe the effect of diffusion on central moments in partial differential equations

(sect. 2). Next, we develop an approach to derive additional reaction terms that may depend on the distribution (sect. 3), expanding beyond previous studies that used raw moments to simulate transit time distributions and only considered radioactive decay. Finally, the central-moment-based method is tested for age distributions and organic matter mineralization in bioturbated sediment (sect. 4) to explore whether a moments-based approach can effectively describe age or reactivity continua in reaction-transport models.

## 2 Derivation of partial differential equations for diffusion

Diffusion is a process that mixes distributions of properties associated with moving particles. In the derivation, we will assume that the property of interest is age, even though it could be any other scalar property that does not affect transport. First, we derive equations for chemical diffusion (see 2.1) and the effect of diffusion on mean age (see 2.2) to illustrate the method based on microscopic diffusion. We then derive partial differential equations for higher centralized moments (see 2.3).

### 2.1 Miscroscopic derivation for concentration

Following Crank (1956), microscopic diffusion can be represented as random jumps forth and back. Consider three locations left (L), center (C), and right (R) aligned on a line and separated by the jumping distance of particles $\delta_x$. The change in the number of molecules at location 'C' is given by

$$\frac{\Delta n_C}{\Delta t} = 0.5 f_r (n_R - n_C) - 0.5 f_l (n_C - n_L) \tag{1}$$

where $f_x$ is the jumping frequency in two directions, and $n_X$ is the number of particles at location X. Smaller case and upper case subscripts indicate evaluations at the boundaries and centers of cells, respectively. After dividing by volume $V$, defining $C = n/V$, and also multiplying the right-hand side by $\delta_x^2/\delta_x^2$

$$\frac{\Delta C_C}{\Delta t} = \frac{1}{\delta_x} \left( \frac{0.5 f_r \delta_x^2 (C_R - C_C)}{\delta_x} - \frac{0.5 f_l \delta_x^2 (C_C - C_L)}{\delta_x} \right) \tag{2}$$

is obtained. The diffusivity is identified as $D = 0.5 f \delta_x^2$. One linearization is made, i.e. $\Delta C = \partial C / \partial x \delta_x$, resulting in

$$\frac{\Delta C_C}{\Delta t} = \frac{1}{\delta_x} \left( D_r \left. \frac{\partial C}{\partial x} \right|_r - D_l \left. \frac{\partial C}{\partial x} \right|_l \right) \tag{3}$$

Applying the divergence theorem yields

$$\frac{\partial C}{\partial t} = \frac{\partial}{\partial x} \left( D \frac{\partial C}{\partial x} \right) \tag{4}$$

which is the diffusion equation.

### 2.2 Microscopic derivation of the diffusion equation for the mean

The same method is applied to the mean age associated with particles, which is the first raw moment. Let $\chi_i$ be the age of a particle and $\sum_{i=1}^{n} \chi_i$ the total age of all particles $n$ in a control volume $V$, so that the mean age of the particles is $\mu = \sum \chi / n$. Then $\mu C = \sum \chi / V$ is the summed ages of all particles per control volume.

Let $j_1$ and $j_4$ be the fluxes that transport particles into the control volume from left and right, respectively. Similarly, let $j_2$ and $j_3$ be the fluxes that remove matter in the rightward and leftward direction, respectively. Substituting the summed total age of particles for the total number of particles in equation 1 yields

$$\frac{\Delta (C_C \mu_C)}{\Delta t} = \frac{1}{V} \left[ j_4 \mu_{j_4} - j_3 \mu_{j_3} - (j_2 \mu_{j_2} - j_1 \mu_{j_1}) \right] \tag{5}$$

**Table 1.** Notation

| | | |
|---|---|---|
| $n$ | number of particles | mol |
| $f$ | particle jumping frequency | T$^{-1}$ |
| $\delta_x$ | particle jumping distance | L |
| $j$ | particle flux | mol T$^{-1}$ |
| $\lambda$ | direction of particle flux | |
| $V$ | volume | L$^3$ |
| $t$ | time | T |
| $C$ | concentration | mol L$^{-3}$ |
| $D$ | diffusivity | L$^2$ T$^{-1}$ |
| $\chi$ | particle property such as age or reactivity | |
| $\mu$ | mean ($\sum \chi / n$) | |
| $\mu_q$ | q-th raw moment ($\sum \chi^q / n$) | |
| $\sigma^2$ | variance ($\sum [\chi - \mu]^2 / n$) | |
| $S$ | skewness ($\sum [\chi - \mu]^3 / n$) | |
| $\phi_q$ | q-th central moment ($\sum [\chi - \mu]^q / n$) | |
| $J$ | diffusive transport terms listed in Table 2 | |
| $R$ | reaction rate | mol L$^{-3}$ T$^{-1}$ |
| $k$ | reactivity | T$^{-1}$ |
| $P$ | producion rate | mol L$^{-3}$ T$^{-1}$ |
| $P_q$ | Production term for q-th moment | |
| $r(\chi)$ | rate expression | |
| $\omega$ | advective velocity | L T$^{-1}$ |
| $g(\chi, \boldsymbol{w})$ | distribution function | |
| $\boldsymbol{w}$ | distribution parameters | |
| $f(\chi)$ | function dependent on distributed property | |
| $v$ | reactivity parameter (see equation 37) | |
| $\alpha$ | reactivity parameter (see equation 37) | T |

whereby $\mu_j$ are the mean ages of the jumping particles, and the fluxes $j_k$ have dimensions of number of particles over time. Note that this section only considers changes to the local mean age caused by diffusive transport. In section 4, derivations will also account for the effect of aging on the mean. From equation 1 follows that $j_1 = 0.5 f_l n_L$, $j_2 = 0.5 f_l n_C$, $j_3 = 0.5 f_r n_C$, and $j_4 = 0.5 f_r n_R$. When it is assumed that the random jumps are not affected by age, the mean age of a larger number of jumping particles will approach the mean age at the source location $X$, so that $\langle \mu_{j_k} \rangle = \mu_X$. Making this substitution and repeating the steps that were taken for the derivation of chemical diffusion yields

$$\left\langle \frac{\Delta(C_C \mu_C)}{\Delta t} \right\rangle = \frac{1}{\delta_x} \left( \frac{D_r(C_R \mu_R - C_C \mu_C)}{\delta_x} - \frac{D_l(C_C \mu_C - C_L \mu_L)}{\delta_x} \right) \tag{6}$$

Again a linearizing assumption $\Delta(\mu C) = \partial(\mu C)/\partial x \delta_x$ is made, after which the partial differential equation

$$\frac{\partial(C\mu)}{\partial t} = \frac{\partial}{\partial x}\left(D\frac{\partial(C\mu)}{\partial x}\right) \tag{7}$$

is obtained. Deleersnijder et al. (2001) derived this equation with a generalized macroscopic approach.

## 2.3 Derivation of partial differential equations for higher centralized moments

Centralized moments are defined as

$$\phi_q = \frac{\sum_{i=1}^{n}(\chi_i - \mu)^q}{n} \tag{8}$$

The zeroth and first centralized moments are always one and zero, respectively. The variance ($\sigma^2$), skewness, and other higher moments correspond to $q = 2$, $q = 3$, and $q > 3$. Throughout the text, we shall refer to raw moments as $\mu_q = n^{-1}\sum \chi_i^q$ and to non-central moments in general as $n^{-1}\sum_{i=1}^{n}(\chi_i - \psi)^q$, where $\psi \neq \mu$.

Considering the exchange of matter with the surroundings through the fluxes $j_k$ (section 2.2), the change of q-powered differences in the control volume can be described by

$$\frac{1}{V}\sum_{i=1}^{n_n}(\chi_i - \mu_o)^q = \frac{1}{V}\sum_{j=1}^{n_o}(\chi_j - \mu_o)^q + \frac{1}{V}(\phi_{j_1}j_1 - \phi_{j_2}j_2 - \phi_{j_3}j_3 + \phi_{j_4}j_4)\Delta t \tag{9}$$

whereby $n_n$ and $n_o$ denote the number of particles in the updated and old population, respectively. All differences from the mean in equation 9, including those associated with mass fluxes, are relative to $\mu_o$. A Taylor series expansion of $\phi$ around $\mu_o$ is used to relate the new state of a population (the left-hand side of equation 9) to the new mean age

$$\frac{1}{V}\sum_{i=1}^{n_n}(\chi_i - \mu_n)^q = \frac{1}{V}\sum_{i=1}^{n_n}(\chi_i - \mu_o)^q + C_n\phi'\Delta\mu + C_n\frac{\phi''}{2}\Delta\mu^2 + C_n\frac{\phi'''}{6}\Delta\mu^3 + \ldots \tag{10}$$

where $C_n = n_n/V$, $\Delta\mu = \mu_n - \mu_o$, and $\phi' = \partial\phi/\partial\mu$, etc. The term on the left-hand side of equation 10 and the first on the right-hand-side of equation 9 can be replaced by $C_n\phi_n$ and $C_o\phi_o$, respectively. By inserting equation 10 into equation 9 and rearranging the terms, the expression

$$C_n\phi_n - C_o\phi_o - C_n\phi'\Delta\mu - C_n\frac{\phi''}{2}\Delta\mu^2 - C_n\frac{\phi'''}{6}\Delta\mu^3 - \ldots = \frac{1}{V}(\sum_{k=1}^{4}\lambda_k\phi_{j_k}j_k)\Delta t \tag{11}$$

is obtained, whereby $\lambda_k = \pm 1$ depending on the direction of the flux.

## 2.3.1 Derivation of a partial differential equation for variance

The derivatives of variance in the Taylor series are

$$\frac{\partial\sigma^2}{\partial\mu} = -2\frac{\sum(\chi_i - \mu)}{n} = 0 \tag{12a}$$

$$\frac{\partial^2\sigma^2}{\partial\mu^2} = 2 \tag{12b}$$

$$\frac{\partial^3\sigma^2}{\partial\mu^3} = 0 \tag{12c}$$

**Table 2.** Partial differentials equation for diffusion of concentration, mean, and centralized moments.

| Moment | Variable | Diffusion equation |
|---|---|---|
| Concentration | $C$ | $\dfrac{\partial C}{\partial t} = \dfrac{\partial}{\partial x}\left(D\dfrac{\partial C}{\partial x}\right)$ |
| Mean | $\mu$ | $\dfrac{\partial(C\mu)}{\partial t} = \dfrac{\partial}{\partial x}\left(D\dfrac{\partial(C\mu)}{\partial x}\right)$ |
| Variance | $\phi_2 = \sigma^2$ | $\dfrac{\partial(C\sigma^2)}{\partial t} = \dfrac{\partial}{\partial x}\left(D\dfrac{\partial(C\sigma^2)}{\partial x}\right) + 2DC\left(\dfrac{\partial\mu}{\partial x}\right)^2$ |
| Higher moments | $\phi_q$ | $\dfrac{\partial(C\phi_q)}{\partial t} = \dfrac{\partial}{\partial x}\left(D\dfrac{\partial(C\phi_q)}{\partial x}\right) + 2qDC\dfrac{\partial\phi_{q-1}}{\partial x}\dfrac{\partial\mu}{\partial x}$ |

See for the definition of the centralized moments ($\phi_q$) equation 8.

The only non-zero derivative is inserted into equation 11. The linearization $\Delta\mu = \partial\mu/\partial t\,\Delta t$ is made, and the result is divided by $\Delta t$. Taking the limit of $\Delta t$ to zero yields

$$\lim_{\Delta t\to 0}\left[\frac{C_n\sigma_n^2 - C_o\sigma_o^2}{\Delta t} - C_n\left(\frac{\partial\mu}{\partial t}\right)^2\Delta t\right] = \frac{1}{V}\sum_{k=1}^{4}\lambda_k\sigma_{j_k}^2 j_k \tag{13}$$

or, under the assumption that $\partial\mu/\partial t$ is finite,

$$\frac{\partial(C\sigma^2)}{\partial t} = \frac{1}{V}\sum_{k=1}^{4}\lambda_k\sigma_{j_k}^2 j_k \tag{14}$$

in differential form.

In the next step, the unknown fluxes on the right-hand side are expressed by known local properties, which can only be done for expected mean values of a large number of random particle jumps. It will be assumed for the partial differential equation for variance, as well as for higher order moments, that i) the flux is determined by the average jumping frequency and the number of particles from a source location X, i.e. $j_k = fn_X$, ii) that q-powered differences reflect the average differences from the location where the particles are jumping, i.e. $\phi_{j_k} = \phi_X$, and iii) that the properties of particles do not effect the jumping probability, i.e. $\langle j_k\phi_{j_k}\rangle = \langle j_k\rangle\langle\phi_{j_k}\rangle$.

With these assumptions, one can write

$$\left\langle\frac{1}{V}\sum_{k=1}^{4}\lambda_k\sigma_{j_k}^2 j_k\right\rangle = \frac{f_r}{2}\left[\sigma_R^2(\mu_C)C_R - \sigma_C^2(\mu_C)C_C\right] - \frac{f_l}{2}\left[\sigma_C^2(\mu_C)C_C - \sigma_L^2(\mu_C)C_L\right] \tag{15}$$

Using the Taylor series for spatial instead of temporal derivatives, i.e. $\Delta\mu = \mu_R - \mu_C$ or $\mu_C - \mu_L$, gives according to equations 12a - 12c

$$\sigma_L^2(\mu_C) = \sigma_L^2(\mu_L) + (\mu_C - \mu_L)^2 \tag{16a}$$
$$\sigma_R^2(\mu_C) = \sigma_R^2(\mu_R) + (\mu_C - \mu_R)^2 \tag{16b}$$

and substituting these into equation 15 yields

$$
\left\langle \frac{1}{V} \sum_{k=1}^{4} \lambda_k \sigma_{j_k}^2 j_k \right\rangle = \frac{f_r}{2} \left[ (\sigma_R^2(\mu_R) + (\mu_C - \mu_R)^2) C_R - \sigma_C^2(\mu_C) C_C \right] -
$$
$$
\frac{f_l}{2} \left[ \sigma_C^2(\mu_C) C_C - (\sigma_L^2(\mu_L) + (\mu_C - \mu_L)^2) C_L \right] \tag{17}
$$

Ignoring the terms with derivatives obtained from the Taylor series for the moment, a part of the equation can be isolated

$$
\left\langle \frac{1}{V} \sum_{k=1}^{4} \lambda_k \sigma_{j_k}^2 j_k \right\rangle^* = \frac{f_r}{2} \left[ \sigma_R^2(\mu_R) C_R - \sigma_C^2(\mu_C) C_C \right] - \frac{f_l}{2} \left[ \sigma_C^2(\mu_C) C_C - \sigma_L^2(\mu_L) C_L \right] \tag{18}
$$

which is similar to equation 5. A linearization of $\partial(\sigma^2 C)/\partial x$ and repeating the procedure leading from equation 5 to equation 7 gives here

$$
\left\langle \frac{1}{V} \sum_{k=1}^{4} \lambda_k \sigma_{j_k}^2 j_k \right\rangle^* = \frac{\partial}{\partial x} \left( D \frac{\partial(C\sigma^2)}{\partial x} \right) \tag{19}
$$

The remaining terms not accounted for yet are

$$
\left\langle \frac{1}{V} \sum_{k=1}^{4} \lambda_k \sigma_{j_k}^2 j_k \right\rangle^{**} = \frac{f_r}{2} (\mu_C - \mu_R)^2 C_R + \frac{f_l}{2} (\mu_C - \mu_L)^2 C_L \tag{20}
$$

which can also be written as

$$
\left\langle \frac{1}{V} \sum_{k=1}^{4} \lambda_k \sigma_{j_k}^2 j_k \right\rangle^{**} = \frac{D_r}{\delta_x^2} \left( \frac{\partial \mu}{\partial x} \delta_x \right)^2 C_R + \frac{D_l}{\delta_x^2} \left( \frac{\partial \mu}{\partial x} \delta_x \right)^2 C_L \tag{21}
$$

yielding in the limit of $\Delta x \to 0$

$$
\left\langle \frac{1}{V} \sum_{k=1}^{4} \lambda_k \sigma_{j_k}^2 j_k \right\rangle^{**} = 2DC \left( \frac{\partial \mu}{\partial x} \right)^2 \tag{22}
$$

Therefore,

$$
\frac{\partial(C\sigma^2)}{\partial x} = \frac{\partial}{\partial x} \left( D \frac{\partial(C\sigma^2)}{\partial x} \right) + 2DC \left( \frac{\partial \mu}{\partial x} \right)^2 \tag{23}
$$

is the final result describing the effect of diffusion on the centralized variance. As demonstrated in supplement section 2.1, the PDE for diffusion of the raw variance can be derived from equation 23 and matches with the result of Delhez and Deleersnijder (2002), which shows that the additional linearizations made in the derivation do not affect the accuracy.

### 2.3.2 Derivation of partial equations for skewness and all higher order moments

For a finite $\partial \mu / \partial t$, dividing equation 11 by an infinitesimally small time step will drop the higher-order terms in the Taylor series, leaving

$$
\frac{\partial(C\phi)}{\partial t} - C\phi' \frac{\partial \mu}{\partial t} = \frac{1}{V} \sum_{k=1}^{4} \lambda_k \phi_{j_k} j_k \tag{24}
$$

The presence of a non-zero first-order derivative makes the derivation of the PDEs for higher-order moments different from that of the variance. It can be found in appendix A and is further analytically validated in the supplement section 2.2. Table 2 shows an overview of all diffusion PDEs.

## 3   Derivation of reaction terms for partial differential equations of moments

Here, we will first give a general mathematical approach to derive reaction terms that can be incorporated in the PDEs for centralized moments (sect. 3.1). The application of this method to particular kinetic expressions relevant to the test applications in this paper will be demonstrated in section 3.2.

### 3.1   General derivation of differential terms for reactions

Reactions change the concentration and can also influence the shape of distributions characterized by their mean and higher

moments. To evaluate the effect of reactions on central moments, we start with an alternative notation for the definition of central moments

$$C\phi_q = \frac{1}{V} \sum_{p=0}^{q} \left[ \binom{q}{p} (-\mu)^p \sum_{i=1}^{n} \chi_i^{q-p} \right] \tag{25}$$

which can be obtained by applying the binomial theorem to equation 8. Next, we differentiate using the product rule to obtain

$$\frac{d(C\phi_q)}{dt} = \sum_{p=0}^{q} \left\{ (-1)^p \binom{q}{p} \left[ \mu^p \frac{d(C\mu_{q-p})}{dt} + \frac{d\mu^p}{dt} C\mu_{q-p} \right] \right\} \tag{26}$$

whereby $\mu_x = C^{-1} \int \chi^x C_\chi \, d\chi$ can be identified as the $x$-th raw moment, and $C\mu_x = \sum \chi^x / V$. When no subscript is given, $\mu = \mu_1$ denotes the mean. The task at hand is to find expressions for all terms when only the concentration and moments are known from a previous time step during a simulation, and a reaction rate is defined, for which we will assume it follows

$$R = \int r(\chi) \, d\chi \tag{27}$$

as a generic rate expression.

Starting with $\mu_{q-p}$ in the second term (eq. 26), we note that raw moments ($\mu_x$) can be obtained by transforming central moments ($\phi$):

$$\mu_x = \sum_{k=0}^{x} \binom{x}{k} \phi_k \mu^{x-k} \tag{28}$$

The second term is further worked out by substituting

$$\frac{d\mu^p}{dt} = p\mu^{p-1} \frac{d\mu}{dt} \tag{29}$$

Solving for $d(C\phi_q)/dt|_R$ (eq. 26) requires expressions for the terms $d(C\mu_{q-p})/dt|_R$ and $d\mu/dt|_R$ (eq. 29). The first one is obtained by integrating

$$\frac{d(C\mu_{q-p})}{dt}\bigg|_R = \int \chi^{q-p} r(\chi) \, d\chi \tag{30}$$

Following the product rule, this derivative also allows

$$\frac{d\mu}{dt}\bigg|_R = \left[\frac{d(C\mu)}{dt}\bigg|_R - \mu R\right] C^{-1} \tag{31}$$

to be solved.

## 3.2  Implementation of reaction kinetics and aging

Representing the production of new material with a zeroth-order kinetic term ($r = 0$) results in $R = P$ as the constant of
integration when treating the integral in equation 27 as indefinite. When $\chi$ represents age, the PDEs for raw moments times
concentration do not need additional terms to account for production ($d[C\mu_q]/dt|_R = 0$ since $\chi = 0$, eq. 30). However, the pro-
duction will decrease the mean age (eq. 31). To incorporate production into the PDEs of variance and higher central moments,
the additional term $P_q$ is obtained by inserting $d(C\mu_q)/dt|_R = 0$ and $d\mu/dt|_R = -\mu P C^{-1}$ into equation 26.

The formulation

$$r = f(\chi)C(\chi) \tag{32}$$

describes first-order reaction kinetics, whereby $f(\chi)$ is the reactivity as a function of the distributed property, and the inte-
grations in equations 27 and 30 are performed over the domain bounded by the scope of the $\chi$ distribution. However, when
$f(\chi) = -k$ is constant, the reaction rate only depends on concentration ($R = -kC$), as is the case for radioactive decay and
simple first-order kinetics. As these reactions do not discriminate with respect to age, the moments will not change (i.e.,
$d\phi_q/dt|_R = d\mu_q/dt|_R = 0$). Consequently, the reaction term becomes $d(C\phi_q)/dt|_R = R\phi_q$, following the product rule.

The distribution may directly represent the reactivity, e.g., $f(\chi) = -\chi$ (see the application in sect. 4.2). When in total $x$
moments are simulated, functions of $\chi^{x+1}$ will have to be integrated (eqns. 30, 32). For simulations whereby the integrations
are numerically performed, this may substantially impact the computation time.

In the final application (sect. 4.3), a hypothetical first-order reaction rate will be considered, whereby the reactivity function
($f[\chi]$) depends on the inverse of $\chi$, which will represent age. Aging affects the mean but not the concentration. Hence, the
product rule implies $d(C\mu)/\partial t = Cd\mu/dt$, whereby $d\mu/dt$ due to aging will be unity provided the same units are used for $\chi$
and $t$. Aging shifts the distribution along the age axis but does not change the shape of the distribution. It does not contribute
to changes in the differences between particle ages and the mean ($d(\chi_i - \mu)^q = 0$), nor does it impact the concentration. This
implies that $d(C\phi_q)/dt$ due to aging is zero.

## 4  Applications

Three applications related to sedimentary environments are presented. For the sake of simplicity and generality, the effect of
sediment properties, such as porosity and tortuosity, on transport will be ignored. Instead, the focus is on adding reactions.

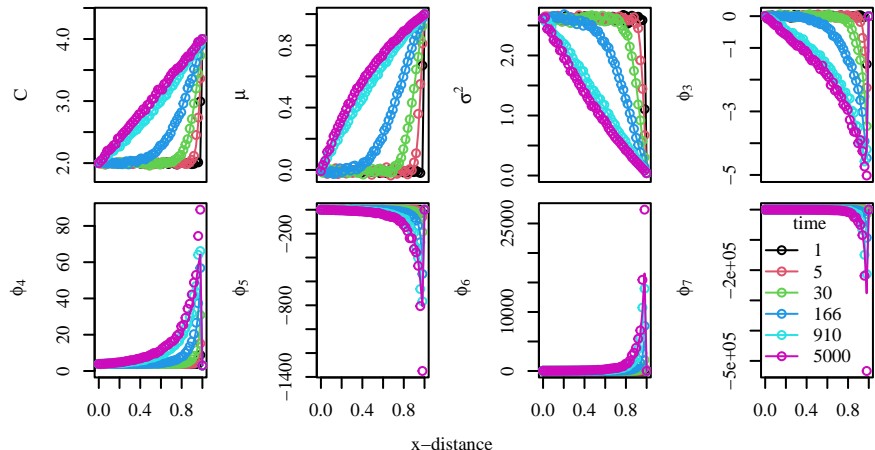

**Figure 1.** Profiles of moments evolving over time due to diffusion. Solution of numerical integration of partial differential equations (solid lines) is compared to a particle tracking simulation (circles). Domain contains 50 cells. The spacing and time steps are set equal to the jumping distance and the inverse jumping frequency of particles, respectively.

### 4.1 Simulating an age tracer

In sediment modeling, resolving the age of a chemical compound understood as the time since its formation or deposition onto the sediment, can help to fit a measured profile and serve other diagnostic purposes.

A general system of equations for an age associated with a chemical concentration $C$ is given by

$$\frac{\partial C}{\partial t} = J_0 - \frac{\partial(\omega C)}{\partial x} + P + R \tag{33a}$$

$$\frac{\partial(C\mu)}{\partial t} = J_1 - \frac{\partial(\omega C\mu)}{\partial x} + C + \mu R \tag{33b}$$

$$\frac{\partial(C\phi_q)}{\partial t} = J_q - \frac{\partial(\omega C\phi_q)}{\partial x} + P_q + \phi_q R \tag{33c}$$

In the equations, $J$ denotes the diffusive transport terms listed in Table 2. The second term accounts for advective transport, whereby $\omega$ is the velocity. In early diagenetic models, the accumulation of the sediment column is typically described as a downward advective burial process, since the sediment surface stays at a zero vertical coordinate. The term $P$ denotes the production of new material with an age of zero, $P_q$ accounts for the effect of production on higher centralized moments, $R$ represents a consumption reaction that does not discriminate with respect to age (e.g., $R = -kC$), and the third term in equation 33b accounts for aging. Refer to section 3.2 for the derivation of these terms.

An example of a simulation involving diffusion without aging, advection, and reactions is shown in Figure 1. Here fixed concentrations and moments were imposed for the last and first cells in the domain as boundary conditions. The initial conditions for the domain were set to the left boundary condition, which has a lower concentration and different moments compared to the right boundary. Over time, there is net chemical diffusion in the leftward direction throughout the domain, eventually

leading to a new steady state. The computed concentration and the first seven moments match well with those computed by a particle-based simulation. There is a small but noticeable mismatch at the peak for the skewness and higher moments, which is potentially due to the finite step size in the Lagrangian simulation.

The Eulerian simulation is based on a finite differences scheme, implemented in R (R Core Team, 2022) and run with the CVODE solver (Brown et al., 1989; Soetaert et al., 2010). The Lagrangian model employed for validating the Eulerian simulation is described in the supplement sect. 1.1. The script to run these simulations is relatively simple and publicly available online. Therein, it is possible to add reactions for production and consumption.

## 4.2 Simulating organic matter mineralization with a reactivity continuum model in turbated sediments

In this application, $C$ denotes organic carbon concentration, $\chi$ is the reactivity (degradation rate coefficient) with dimensions $T^{-1}$, and no explicit aging process is involved. As an example, we will consider the deposition of organic carbon with an initially uniform distribution for $\chi \in [0, m]$, which is described by the state variables concentration, mean reactivity, and variance of the reactivity. The rate expression from equation 32 is applied with $f(\chi) = -\chi$ so that $R = -\mu C$. By working out eq. 26 with these definitions, the following equations

$$\left.\frac{\partial C}{\partial t}\right|_R = -\mu C \tag{34a}$$

$$\left.\frac{\partial (C\mu)}{\partial t}\right|_R = -\int_0^m (\chi^2 C)\, d\chi \tag{34b}$$

$$\left.\frac{\partial (C\sigma^2)}{\partial t}\right|_R = -\int_0^m (\chi^3 C)\, d\chi + 2\mu \int_0^m (\chi^2 C)\, d\chi - \mu^3 C \tag{34c}$$

can be obtained.

The integrals will be evaluated numerically, meaning that the full distribution needs to be constructed from the moments. Based on a finite number of moments, it can only be estimated, and we chose the function

$$g(\chi, \boldsymbol{w}) = C_0 e^{w_0 \sqrt{\chi} + w_1 \chi} + w_2 \chi e^\chi \tag{35}$$

to represent the reactivity distribution. It is motivated as follows: The concentration of unreactive organic matter at the intercept does not change ($g(0, \boldsymbol{w}) = C_0$). For the diffusion-reaction equation $\partial C/\partial t = D \partial^2 C/\partial x^2 - \chi C$, the general solution is $A e^{\pm \sqrt{\chi/D}x}$. The solution for $\partial C/\partial t = -\chi C$ is $C(t) = A e^{-\chi t}$. The first term can capture both these dynamics. The last term is linearly independent and introduces a third fitting parameter to match the number of equations. These terms have the desirable

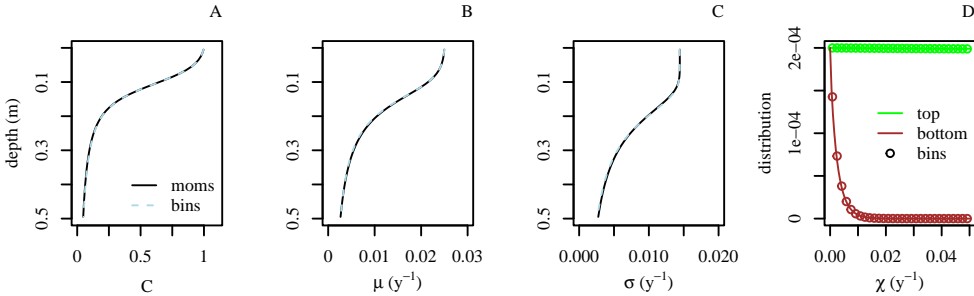

**Figure 2.** A simulated reactivity continuum model run to a steady state (black line), validated against a model using 30 bins to encompass the reactivity range (dashed light blue line). Concentration, mean reactivity, and standard deviation are shown in panels A, B, and C, respectively; panel D depicts the distribution function for reactivities at the upper (green line) and lower (brown line) boundaries of the model domain in comparison to the validation simulation (points).

properties that they cannot fluctuate or become negative and can be evaluated at $\chi = 0$. The equations

$$C - \int_0^m g(\chi, \boldsymbol{w})\, d\chi = 0 \tag{36a}$$

$$\mu - \int_0^m g(\chi, \boldsymbol{w})\chi\, d\chi = 0 \tag{36b}$$

$$\sigma^2 - \int_0^m (\chi - \mu)^2 g(\chi, \boldsymbol{w})\, d\chi = 0 \tag{36c}$$

are solved with a multidimensional root-finding procedure (Soetaert, 2009) to fit the parameter vector $\boldsymbol{w}$.

In the example shown in Figure 2, transport involves bioturbation and advection. The burial velocity was set to $1 \mathrm{~mm\,y^{-1}}$. The bioturbation coefficient had a maximum value of $10^{-10} \mathrm{~m^2\,s^{-1}}$ at the sediment-water interface and decreased exponentially over depth with an e-folding distance set to $2 \mathrm{~cm}$. The uniform distribution implemented as Dirichlet upper boundary

conditions was defined by the moments $C = 1$, $\mu = 2.5 \cdot 10^{-2} \mathrm{~y^{-1}}$, and $\sigma = 1.44 \cdot 10^{-2} \mathrm{~y^{-1}}$, which were also used as initial conditions throughout the domain. A no-gradient condition was set as lower boundary condition. In total, 50 evenly spaced cells discretized a domain length of $50 \mathrm{~cm}$. The simulation, using finite differences (Soetaert and Meysman, 2012), was run with the VODE solver (Brown et al., 1989).

The organic matter concentration imposed to a fixed value at the upper boundary condition decreases due to degradation over

270 depth (Figure 2A). Due to mixing, the mean reactivity of organic matter remains relatively constant in the bioturbated zone but decreases below it (Figure 2B) as the more reactive organic matter is degraded. The variance is also kept relatively stable within the bioturbated zone and decreases strongly below (Figure 2C), as the removal of more reactive material decreases the spreading of the reactivity distribution. The distributions at the top and bottom are shown as well (Figure 2D). The obtained

results closely resemble those of a 30-G model, which partitions the reactivity range defined at the upper boundary into 30 equally spaced distinct reactivity values, treating each bin as an independent state variable (see for more details section 1.2 of the supplementary materials).

## 4.3 Apparent organic matter reactivity as a function of age

In this application, the age distribution is modeled and determines the reactivity of organic carbon. The transport equations are the same as in the previous applications, and aging is included (see eq. 33b). The age-dependent reactivity (eq. 32) is specified as

$$f(\chi) = -\frac{v}{\alpha + \chi} \tag{37}$$

whereby $\alpha$ prevents division by zero and may also be used as a fitting parameter. It resembles the long-established expression for the mean reactivity, $\bar{k} = v/(\alpha + \mu)^\beta$ (Middelburg, 1989). Conceptually it may be the most simple expression to model the effect of aging on reactivity.

Here we consider the application of four moments: concentration ($C$), mean age ($\mu$), variance ($\sigma^2 = \phi_2$), and skewness ($S = \phi_3$). For this, three-parameter distributions describe the distribution's shape and mean, and a fourth parameter serves as a multiplier to adjust the concentration.

We present our analysis using two distinct distributions to represent age continua, which will be compared later to evaluate the distribution shape's role. The first one is the triangular distribution,

$$g(\chi, \boldsymbol{w}) = \begin{cases} w_1 \frac{2(\chi - w_2)}{(w_3 - w_2)(w_4 - w_2)} & \text{if } w_2 \leq \chi \leq w_4 \\ w_1 \frac{2(w_3 - \chi)}{(w_3 - w_2)(w_3 - w_4)} & \text{if } w_4 < \chi \leq w_3 \end{cases} \tag{38}$$

whereby $w_1$ is the multiplier, $w_2$ and $w_3$ denote the lower and upper limit of the distribution (outside this interval, the function evaluates to zero), and $w_4$ corresponds to the mode. Given the closed-form expressions for the central moments of the triangular distribution (Forbes et al., 2010), the distribution parameters for given moments are found by first determining $b$ and $c$ in the equations

$$\sigma^2 - (b^2 + c^2 - bc)/18 = 0 \tag{39a}$$

$$S + (b - 2c)(c - 2b)(b + c)/270 = 0 \tag{39b}$$

with a root-solver, which allows the distribution parameters to be calculated as follows: $w_1 = C$, $w_2 = (3\mu - b - c)/3$, $w_3 = w_2 + b$, and $w_4 = w_2 + c$.

The other demonstrated distribution is the translated Weibull distribution, formulated as

$$g(\chi, \boldsymbol{w}) = \frac{w_1 w_2}{w_3} \left( \frac{\chi - w_4}{w_3} \right)^{w_2 - 1} e^{-\left( \frac{\chi - w_4}{w_3} \right)^{w_2}} \tag{40}$$

whereby $w_1$ serves as a multiplier to adjust the concentration, $w_2$ is the shape parameter, $w_3$ is the scaling parameter, and $w_4$ is the location parameter (Forbes et al., 2010). All parameters were obtained by solving a set of equations as shown in equations 36, along with an additional equation to account for skewness.

The numerical model calculates first the reaction rates for all moments, as described in section 3.2. The PDEs are solved with an implicit finite volume scheme with hybrid differences to account for advection and diffusion, using the implementation from JurRTM (Rooze et al., 2020; Zindorf et al., 2021). For the state variable $C\mu$, the aging term is added to the reaction term. Also, the last term in the diffusion equations for variance and skewness (Table 2) is accounted for by calculating first the $\partial\mu/\partial x$ gradient and adding the resulting term as a reaction rate.

The model divides a domain length of $10$ cm into 50 evenly spaced cells. The upper boundary condition is added as a Dirichlet boundary condition with a prescribed distribution. The initial values are set to the values of the upper boundary condition. A zero-gradient condition is imposed at the lower boundary of the domain. The reactivity parameters (eq. 37) were set to $v = 0.8$ and $\alpha = 1$ y.

The simulations were run for 53 years with a maximum time step of 1 year. For root-solving the 'nleqslv' and 'numDeriv' packages in R were utilized (Hasselman, 2023; Gilbert and Varadhan, 2016; R Core Team, 2022). Numerical integrations were carried out over the domain $\chi \in [0, 150]$. To validate the simulation, a complementary model capturing concentration evolution across binned ages was employed. This validation model adopts matching boundary and initial conditions, along with a congruent simulation setup. Detailed technical documentation for the validation model is provided in the supplementary material (sect. 1.3).

For the simulations shown in Figure 3, the burial velocity was set to $2$ mm y$^{-1}$. The bioturbation coefficient had a maximum value of $10^{-11}$ m$^2$ s$^{-1}$ at the top. It decreased exponentially with depth, having an e-folding distance set to 3 cm, implying that the diffusivity at the bottom of the domain is effectively zero and that the zero-gradient condition has a negligible effect on the results. The distributions set as upper boundary and initial conditions had moments set to $C = 1$, $\mu = 15$ y, $\sigma^2 = 13$ y$^2$, $S = 0$ y$^3$.

The results (Fig. 3) demonstrate accurate concentration and mean age computation throughout the simulations. However, a noticeable mismatch appears in the simulated variance and skewness after 50 years in the triangular distribution simulation, whereas the translated Weibull distribution simulation reproduces the moments accurately.

In the two lower rows of Figure 3, distributions are presented as resolved during simulations for integration. After 15 years, the moment-based simulation distributions closely match the validation simulation, except at 2.5 cm depth for the triangular distribution. Mixing of younger and older materials from the upper boundary results in a positive skew near the upper boundary and a negative skew at greater depths.

Over time, the influence of the initial conditions diminishes. Considering only advection, the time to transport all initial material out of the model domain is 50 years. However, upward diffusion, despite net downward chemical diffusion, can increase the residence time of some particles. The triangular distribution struggles to accurately depict the resulting positively skewed distributions, as it degenerates into nearly right triangles at and below 2.5 cm depth (Fig. 3). The much more versatile Weibull distribution provides a better representation of the age distribution, but the distribution is clearly off in the most actively turbated zone (Fig. 3), which, however, does not appear to affect the accuracy of the simulated moments. When the simulation is run for 100 years (not shown), the moments maintain their accuracy, and the visual comparison of the distribution even slightly improves.

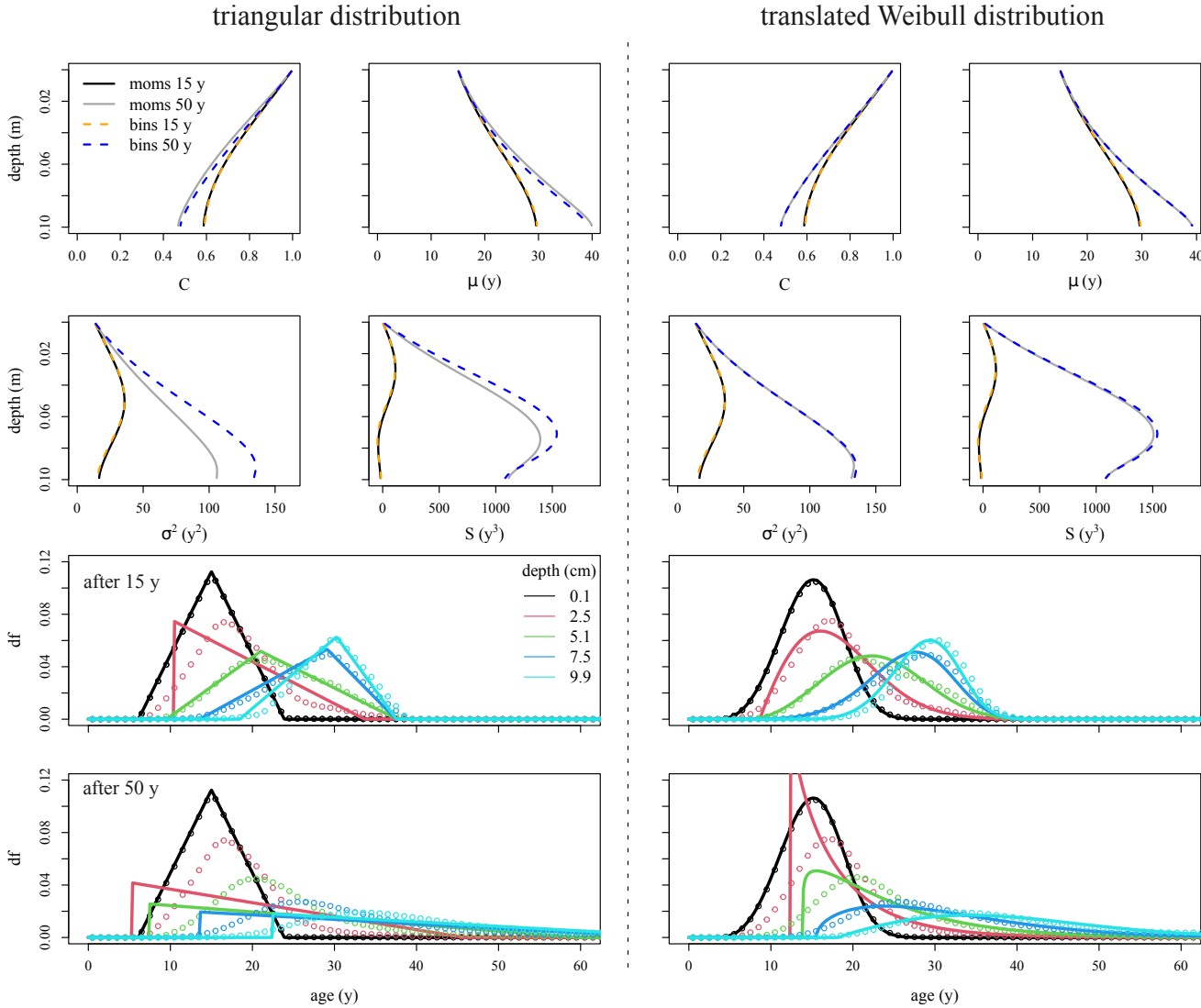

**Figure 3.** Results of simulations based on age distributions. The left and right halves of the figure depict simulations utilizing the triangular and translated Weibull distributions, respectively. Presented are results after both 15 and 50 years of simulation. The moments-based simulation is compared to a simulation using age bins. The moment-based and age bin simulations are distinguished by solid and dashed lines in the upper two rows (see legend) and by lines and points in the lower two rows, which display distribution functions (df). For additional details on the simulations, refer to section 4.3.

Interestingly, the limited impact of the precise shape of the distribution also transpires from the great similarity of the
distributions that evolve at depth over time in the validation simulations, regardless of the imposed distribution type at the upper boundary condition (compare distributions below 2.5 cm depth after 50 years emerging from triangular and Weibull

distributions in Fig. 3). Similar distributions also formed when the reaction was turned off (not shown). Therefore, the interplay between bioturbation and aging appeared to be of greater consequence for the evolution of the moments than the reaction and choice of distribution at the upper boundary.

## 5 Discussion

### 5.1 Evaluation of the applications

The theory outlined in this paper gives modelers a free hand to simulate properties associated with particles described as a concentration. Modelers can use different reaction kinetics and simulate the effect on a continuous distribution. The first application shows that it is possible to reliably simulate numerous moments of an age distribution. In this type of application, it is unnecessary to reconstruct the distribution function from its moments during the simulation. In principle, any distribution is uniquely defined by a large or infinite number of moments. However, in practice, there does not exist a universal solution to retrieve distributions from their moments. Numerical methods have been developed for this purpose (e.g., John et al., 2007; Arbel et al., 2016) but do not always succeed.

In the two other applications, the shape of the distribution affected the reaction rates and, therefore, needed to be resolved in the simulations. In the reactivity continuum model, the shape could be well predicted, i.e., the chosen class of distribution function was a good approximation of the exact solutions. The bounds of the initial distribution of deposited material also provided bounds for the numerical integration of deeper (older) material since the domain only becomes smaller as the more reactive materials are consumed, and the refractory organics remain. These simulations were stable and ran relatively fast, despite the requirement to determine the parameters of the distribution in run-time, which involves several numerical integrations. This approach could be attractive as an alternative to the multi-G approach, as it does not require the arbitrary definition of various reactivity classes (Jørgensen, 1978) and can better represent slight differences in reactivity naturally developing over depth. However, the multi-G simulation will run faster, even though the numerical scheme adopted here could be significantly improved (see suggestions below). The initial uniform distribution used in the simulation may not be realistic for organic matter in marine sediment (Boudreau and Ruddick, 1991). Instead, distributions similar to multi-G or continuous distributions found in the literature (Arndt et al., 2013) can be imposed for freshly deposited organics.

The third application posed the greatest challenge, as the shape of the age distribution strongly changed during the simulation and was hard to predict. The emergence of pronounced asymmetry motivated the addition of skewness as a state variable. The translated Weibull distribution performed best due to its versatility, as it can represent the exponential, Rayleigh, normal, and other two-parameter distributions.

Occasional substantial deviations were observed between distributions reconstructed from their moments and the actual distributions. An illustration of such discrepancies is evident in the left tails of the distributions after 50 years at a depth of 2.5 cm (Figure 3), even though the moments of both the triangular and translated Weibull distributions were nearly or entirely accurate. A fundamental problem of reconstructing distributions from moments pertains to the unequal weighting for concentration differences along the age/reactivity axis. Central moments exhibit zero weight at $\chi = \mu$ (eq. 8), while weights are

maximized at the distribution's extremes. Consequently, the reconstruction process tends to fit the tails better than the central region encompassing the mean and mode. This tendency could introduce biases, particularly in scenarios where the extremes influence the dynamics less; for instance, when the precise age of very old material has minimal impact on overall reactivity. The bias, depending on $q$ in eq. 8, will be stronger for higher-order moments.

Numerous optimizations can be considered to improve the numerical schemes. In the third simulation, a transition from the method of lines (Sarmin and Chudov, 1963), characterized by continuous time and discrete space, to the classical finite differences/volumes approach, characterized by discrete space and time, yielded substantial improvements in simulation time and stability. This approach, affording greater control over time stepping and the execution frequency of root-solving procedures and numerical integrations, could likely also shorten computation times in the other applications. Other polynomial type or spline expressions could be tried to describe the distributions. Finally, pre-calculated search tables for distribution functions could be designed to look up parameters corresponding to a combination of moments. Then numerical integrations during run-time would become obsolete, letting simulations run faster.

## 5.2 The application of central moments-based models in comparison to alternative approaches

Instead of utilizing central moments, an alternative consideration involves using raw moments. When focusing solely on production processes and age distributions are simulated, raw moments could prove more practical, as they are not affected by the production of new material, but the disadvantage will be that aging will still affect them.

In scenarios involving consumption reactions, the use of raw moments generally leads to the evaluation of the fewest number of terms. For example, consider the rate expression in equation 30 compared to those in equation 34 for central moments. However, the steps outlined in section 3.1 can be automated to obtain all necessary terms. For the complete set of equations encompassing all moments, the same integrations must be carried out, regardless of whether raw or central moments are employed. Hence, the choice between raw and central moments may have limited practical significance when considering only the consumption reactions in numerical models.

Central moments can be converted into raw moments (as in eq. 28) or any other non-central moment. This implies that the choice of moment type for the PDEs is not critical for additional steps, such as the reconstruction of distributions from the moments (see previous section).

Central moments have advantages in simulating age or transit time distributions, as it lets aging only affect the mean and not higher moments like variance and skewness. In contrast, PDEs for non-central moments necessitate additional terms to account for aging (Delhez and Deleersnijder, 2002). While moments-based distributions have been employed for simulating age tracers and radioactive decay, their application in more complex dynamics remains unexplored. Determining the practical benefits of central versus non-central moments would require further testing.

For each application within this study, alternative numerical approaches were presented. One such approach involves utilizing Lagrangian simulation, which typically demands more computation time and may be less suitable for boundary value problems involving extended simulation durations needed to reach a steady state. In the second and third applications, continuous distributions were discretized by multiple state variables. While multi-G models run faster and are easier to implement,

continuous approaches become particularly relevant when the goal is to examine aging processes. Discretizing age distributions presents challenges, as aging involves exchanges between different age bins. Similar to numerical advection schemes, this can lead to numerical diffusion, distorting the variance and skewness (Klingbeil et al., 2014). To circumvent this concern, the validation simulation in the third application utilized a moving grid for age bins (supplement sect. 1.3). This approach reaches high accuracy. However, it has the disadvantage of requiring many state variables to represent age classes for the simulated period, which could become problematic, particularly in simulations with larger grids. Moments-based simulations offer an elegant and efficient solution, while the versatile applicability of PDEs makes their implementation in various models more convenient.

## 6 Conclusions

The derived diffusion-advection-reaction PDEs for central moments can be valuable tools for assessing the effect of processes on distributions, computing transit time/age distributions, and simulating reactivity distributions. Central moments hold advantages over raw moments, being intuitively interpretable and unaffected by aging.

The central moments of transit time/age distributions can be simulated first to allow the actual distribution reconstruction afterward. When the reactivity depends on the distribution, the distribution must be reconstructed and integrated at each time step. An adequate function could be defined to carry out the reconstruction from the mean and variance for the simulated distributions representing the reactivity continua in the second application. However, resolving age distributions in the third application to compute reactivities based on ages encountered distribution-choice-related accuracy challenges, suggesting a need for additional validation, particularly when the distribution is more sensitive to the reaction term and vice versa.

The second and third applications underscored bioturbation's considerable influence on chemicals' reactivity and age distributions in surficial sediments, highlighting potential inaccuracies in prior reactivity continuum approaches that ignore mixing. Despite employing realistic transport parameters, applying the models to field data remains essential, particularly for more robust validation of the chosen reaction dynamics. Also, a more thorough analysis is required to assess the significance of age/reactivity distribution shapes for mineralization rates within mixed zones. The framework developed within this study is well-suited to address these aspects in future research.

*Code availability.* The scripts for the applications are available at https://git.io-warnemuende.de/rooze/DiffussionCentralMoments/src/tag/2.0.

## Appendix A: Derivation of diffusion PDEs for higher centralized moments

Continuing the derivation of the PDE for the diffusion of higher moments from equation 24, one can write

$$\frac{1}{V}\left\langle \sum \lambda_k \phi_{j_k} j_k \right\rangle = -\frac{f_l}{2}\left[\phi_C(\mu_C)C_C - \phi_L(\mu_C)C_L\right] + \frac{f_r}{2}\left[\phi_R(\mu_C)C_R - \phi_C(\mu_C)C_C\right] \tag{A1}$$

when the same assumptions are made with regard to the fluxes as in the derivation for the PDE of the variance. The fluxes $j_1$ and $j_4$, transporting material into the control volume, can be written as functions of the mean age at the source location,

$$\phi_L(\mu_L) = \phi_L(\mu_C) + \phi' \cdot (\mu_L - \mu_C) + 0.5\phi'' \cdot (\mu_L - \mu_C)^2 + \ldots \tag{A2a}$$

$$\phi_R(\mu_R) = \phi_R(\mu_C) + \phi' \cdot (\mu_R - \mu_C) + 0.5\phi'' \cdot (\mu_R - \mu_C)^2 + \ldots \tag{A2b}$$

Inserting these equations into equation A1, the part not accounting for the derivatives of the Taylor series is isolated

$$\frac{1}{V}\left\langle \sum \lambda_k \phi_{j_k} j_k \right\rangle^* = \frac{\partial}{\partial x}\left( D \frac{\partial(C\phi)}{\partial x} \right) \tag{A3}$$

whereby $\partial(C\phi)/\partial x$ has been linearized. The terms for the derivatives can be written as

$$\frac{1}{V}\left\langle \sum \lambda_k \phi_{j_k} j_k \right\rangle^{**} = -\frac{f_l C_L}{2}\left( \frac{\partial\phi}{\partial\mu}\bigg|_L (\mu_L - \mu_C) + \frac{1}{2}\frac{\partial^2\phi}{\partial\mu^2}\bigg|_L (\mu_L - \mu_C)^2 + \ldots \right) -$$
$$\frac{f_r C_R}{2}\left( \frac{\partial\phi}{\partial\mu}\bigg|_R (\mu_R - \mu_C) + \frac{1}{2}\frac{\partial^2\phi}{\partial\mu^2}\bigg|_R (\mu_R - \mu_C)^2 + \ldots \right) \tag{A4}$$

Substituting this and $f/2 = D/\delta_x^2$ into the last equation yields

$$\frac{1}{V}\left\langle \sum \lambda_k \phi_{j_k} j_k \right\rangle^{**} = \frac{D_l C_L}{\delta_x}\left( \frac{\partial\phi}{\partial\mu}\bigg|_L \frac{\Delta\mu}{\delta_x}\bigg|_l - \frac{1}{2}\frac{\partial^2\phi}{\partial\mu^2}\bigg|_L \frac{(\Delta\mu)^2}{\delta_x}\bigg|_l + \ldots \right) - \frac{D_r C_R}{\delta_x}\left( \frac{\partial\phi}{\partial\mu}\bigg|_R \frac{\Delta\mu}{\delta_x}\bigg|_r + \frac{1}{2}\frac{\partial^2\phi}{\partial\mu^2}\bigg|_R \frac{(\Delta\mu)^2}{\delta_x}\bigg|_r + \ldots \right) \tag{A5}$$

whereby $\mu_C - \mu_L = \Delta\mu_l$ and $\mu_R - \mu_C = \Delta\mu_r$. Taking the limit of $\Delta x$ to zero, the second-order Taylor series terms will drop. Linearizing $\partial\mu/\partial x$ yields

$$\frac{1}{V}\left\langle \sum \lambda_k \phi_{j_k} j_k \right\rangle^{**} = \frac{D_l C_L}{\delta_x}\frac{\partial\phi}{\partial\mu}\bigg|_L \frac{\partial\mu}{\partial x}\bigg|_l - \frac{D_r C_R}{\delta_x}\frac{\partial\phi}{\partial\mu}\bigg|_R \frac{\partial\mu}{\partial x}\bigg|_r \tag{A6}$$

Inserting the linearizations

$$\frac{\partial\phi}{\partial\mu}\bigg|_L = \frac{\partial\phi}{\partial\mu}\bigg|_C - \frac{\partial^2\phi}{\partial\mu\partial x}\bigg|_C \delta_x \tag{A7a}$$

$$\frac{\partial\phi}{\partial\mu}\bigg|_R = \frac{\partial\phi}{\partial\mu}\bigg|_C + \frac{\partial^2\phi}{\partial\mu\partial x}\bigg|_C \delta_x \tag{A7b}$$

into equation A6 gives

$$\frac{1}{V}\left\langle \sum \lambda_k \phi_{j_k} j_k \right\rangle^{**} = \frac{\partial\phi}{\partial\mu}\left( \frac{D_l C_L}{\delta_x}\frac{\partial\mu}{\partial x}\bigg|_l - \frac{D_r C_R}{\delta_x}\frac{\partial\mu}{\partial x}\bigg|_r \right) - 2DC\frac{\partial}{\partial x}\left( \frac{\partial\phi}{\partial\mu} \right)\frac{\partial\mu}{\partial x} \tag{A8}$$

The concentration gradient is also linearized

$$C_L = C_l - \frac{1}{2}\frac{\partial C}{\partial x}\delta_x \tag{A9a}$$

$$C_R = C_r + \frac{1}{2}\frac{\partial C}{\partial x}\delta_x \tag{A9b}$$

When these expressions are inserted, and the limit of $\delta_x$ to zero is taken, the following partial differential equation

$$\frac{1}{V}\left\langle \sum \lambda_k \phi_{j_k} j_k \right\rangle^{**} = -\frac{\partial\phi}{\partial\mu}\left[ \frac{\partial}{\partial x}\left( DC\frac{\partial\mu}{\partial x} \right) + D\frac{\partial C}{\partial x}\frac{\partial\mu}{\partial x} \right] - 2DC\frac{\partial}{\partial x}\left( \frac{\partial\phi}{\partial\mu} \right)\frac{\partial\mu}{\partial x} \tag{A10}$$

is obtained.

In the remaining steps, it will be shown that the second term on the left-hand side of equation 24 will cancel out with the first term on the right-hand side of equation A10. Applying the product rule to $\partial(C\mu)/\partial t$ and substituting $\partial C/\partial t$ and $\partial(C\mu)/\partial t$ with equations 4 and 7 results in

$$C\frac{\partial\mu}{\partial t} = \frac{\partial}{\partial x}\left(D\frac{\partial(C\mu)}{\partial x}\right) - \mu\frac{\partial}{\partial x}\left(D\frac{\partial C}{\partial x}\right) \tag{A11}$$

Since the product rule, applied twice, also implies

$$\frac{\partial}{\partial x}\left(D\frac{\partial(C\mu)}{\partial x}\right) = \frac{\partial}{\partial x}\left(DC\frac{\partial\mu}{\partial x}\right) + \mu\frac{\partial}{\partial x}\left(D\frac{\partial C}{\partial x}\right) + D\frac{\partial C}{\partial x}\frac{\partial\mu}{\partial x} \tag{A12}$$

equation A11 can be recast into

$$C\frac{\partial\mu}{\partial t} = \frac{\partial}{\partial x}\left(DC\frac{\partial\mu}{\partial x}\right) + D\frac{\partial\mu}{\partial x}\frac{\partial C}{\partial x} \tag{A13}$$

which matches the part between square brackets in equation A10. These terms will cancel each other out when the last equation is inserted on the left-hand side and equations A3 and A10 on the right-hand side of equation 24, leaving

$$\frac{\partial(C\phi)}{\partial t} = \frac{\partial}{\partial x}\left(D\frac{\partial(C\phi)}{\partial x}\right) - 2DC\frac{\partial}{\partial x}\left(\frac{\partial\phi}{\partial\mu}\right)\frac{\partial\mu}{\partial x} \tag{A14}$$

Finally, by substituting

$$\frac{\partial\phi_q}{\partial\mu} = -q\phi_{q-1} \tag{A15}$$

the final result shown in Table 2 is derived.

*Author contributions.* The study was conceived and designed by J.R. in collaboration with H.J. and H.R. All authors discussed the results and commented on the manuscript.

*Competing interests.* The authors declare that they have no competing interests.

*Acknowledgements.* We thank the anonymous reviewers for their constructive criticism and insightful comments. This work was conducted within the DAM pilot mission "MGF-Ostsee" (Grant No. 03F0848A) funded by the German Federal Ministry of Education and Research. Heewon Jung was supported by the National Research Foundation of Korea (NRF) grant funded by the Korea government (MSIT) (No. 2022R1C1C1004512).

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
