# Peer review of "A Novel Eulerian Model Based on Central Moments to Simulate Age and Reactivity Continua Interacting with Mixing Processes"

_EGUsphere, 2023_

## Author Response (AR1)

Dear reviewers,

We are very thankful for the time invested in reviewing our manuscript and the constructive comments and suggestions.

After considering all the comments, we have made the following additions:
1) a section with a theoretical derivation for reaction terms (see section 3 in the original manuscript), placed before the section with the applications (section 4),
2) subsection 5.2 in the discussion, which elaborates on the practical merits of using models based on central moments compared to alternatives, including non-central moments,
3) a table that lists the variables in the manuscript and explains their meaning (Table 1),
4) use of alternative distributions in the third application (see equations 38 and 40),
5) improvements to the introduction.

Also, a supplement is added, which provides
1) a technical description of the numerical models used for testing the applications in a supplement (see supplement section 1),
2) added proof for the validity of the diffusion equation for variance (supplement section 2.1),
3) added analytical validation for the higher-moment equation, using a delta distribution in an initial value problem (supplement section 2.2).

We have significantly improved the third application. As a result, Fig. 3 has changed. This figure also shows how distribution evolves at several depths. It made Fig. 4 somewhat redundant, which was, therefore, removed. The corresponding discussion was also changed accordingly, and we edited the abstract.

Following reviewer suggestions, section 5.2 was added to compare the approach based on central moments to other approaches.

Also, we decided to change the title to let it include "central moments." We deleted Reaction-Transport, as this is already implied by "reactivity continua" and "mixing processes," to make the title shorter.

Below, we provide a more detailed response to the reviewer's suggestions.

During the review process the code can be downloaded from https://drive.google.com/drive/folders/1UdQuAaxXq-VTA1JKyMD_7DDIy5HRJCal?usp=sharing
We plan to add a repository with a doi after the review process if the manuscript is accepted.

**Response to reviewer 1**

**Original comment:** The authors presented formulation of diffusive transfers of materials with central moments in detail. In its application to OM early diagenesis the authors also introduced production/consumption of central moments focusing on age and OM reactivity though in less detail and less organized way in my opinion. Overall, new approaches are very intriguing in that they may require less assumptions on the continuum and have a potential of more flexibility to track variable properties of interests in materials important for early diagenesis in marine sediments including age and reactivity. My concern is that I had some difficulty in following the formulation

of production/consumption of moments and comparison of the features of the new approach to those from others adopting noncentral moments might be insufficient. If the difficult formulation of the production terms is actually a feature of central moments compared to noncentral moments, I would like the authors to provide more detailed explanations/formulations/comparison. Otherwise, I think this paper is suited for GMD.

**Answer:** We are glad to read that the reviewer finds the new approaches intriguing and thank him/her for his/her constructive remarks. Regarding the formulation of production/consumption terms, he/she raises the concern that a) the derivations are hard to follow and b) that the formulation of reaction/production terms for central moments and non-central moments are not compared and discussed.

Concerning point 'a': In the new section "*3. Derivation of reaction terms for partial differential equations of moments,*" the first subsection "*3.1 General derivation of differential terms for reactions*" explains how the reaction terms, in general, can be systematically derived. However, the precise formulation will depend on the reaction type and kinetics. Section "*3.2 Implementation of reaction kinetics and aging*" shows how to apply this method to specific reactions considered in the applications.

Regarding point 'b': The reaction rate term for raw moments is given by equation 29. Central moments indeed need additional terms to account for changes in the mean. However, we would like to stress that this will practically not matter in numerical simulations, as the steps described in section 3 can be programmed. For the set of PDEs of central and raw moments, the same integrations have to be performed.

There are only specific reactions/processes that may favor the use of either raw or central moments. When the age is modeled, and only production terms are considered, raw moments may be preferable, as the production of new material with an age of zero will not affect $C\mu_q$ (an increase in C and a decrease in $\mu_q$ cancel a change in $C\mu_q$ out). The aging process favors central moments, as they are not affected. This is now explained in the new subsection in the discussion "*5.2 The application of central-based models in comparison to alternative approaches.*"

**Original comment:** Definition of variables is sometimes confusing. Multiple uses of X, µ, w, k, j for different variables, for instance.

**Answer:** We have addressed this by introducing a variable glossary in Table 1 and revising the content to prevent symbol ambiguity.

**Original comment:** L163-166. Production of total age in Eq. 25b makes sense to me but is there any way to verify this formulation?

**Answer:** This term can be obtained by applying the product rule: $\partial_t(C\mu) = C\partial_t\mu + \mu\partial_t C$. Aging does not affect the concentration ($\partial_t C = 0$) and aging is proportional to change in time ($d\mu = dt$ implying $\partial_t\mu = 1$), so that $\partial_t(C\mu) = C$. This is now explained in section 3.2 (lines 208-212: "Aging affects... is zero.").

**Original comment:** L169. I assume tau is defined as age but did the authors already define the symbol before? If this is the case, then is C necessary before tau_i on the right-hand side of Eq. 26?

**Answer:** In the revised manuscript we changed both $\tau$ and X to $\chi$.

In equation 25 (eq. 26 in the original manuscript) does not require C before $\chi_i$ (previously $\tau_i$). Here is a short derivation. The binomial theorem implies

$$(\chi - \mu)^q = \sum_{j=0}^{q} \binom{q}{j} \chi^{q-j} (-\mu)^j$$

Using the expected value notation, we have

$$\langle(\chi - \mu)^q\rangle = \sum_{j=0}^{q} \binom{q}{j} \langle\chi^{q-j}\rangle (-\mu)^j$$

and

$$\langle(\chi - \mu)^q\rangle = \sum_{j=0}^{q} \binom{q}{j} \frac{\sum \chi^{q-j}}{n} (-\mu)^j$$

Given that $\phi_q = \langle(\chi - \mu)^n\rangle$, multiplication by the number of particles $n$ in a control volume gives

$$C\phi_q = \sum_{j=0}^{q} \binom{q}{j} \sum \chi^{q-j} (-\mu)^j$$

which is equation 25.

**Original comment:** Section 3.1. More explanation would be helpful in general. Particularly, derivation of production terms in Eqs. 25b and c. For instance, some examples with specified q value would be helpful, or some more explanation with example when switching tracer from age to reactivity in the next section (Section 3.2)? Also, it may be interesting to have some insight into the difference in formulation of production/consumption terms from previous studies especially those adopting noncentral moments (e.g., Delhez and Deleersnijder, 2002). I assume the central moments may have more complicated formulation for production/consumption compared to non-central counterparts in general? If this is actually a general feature of central moments, this should be described.

**Answer:** We agree with the reviewer that the derivations should have been clearer. Therefore, we added the new section 3. Writing out binomial coefficients for specific examples by hand will take up too much space and is therefore not included in the manuscript. Delhez and Deleersnijder (2002) considered aging and radioactive decay but not reaction processes with kinetics that depend on the distributions of the property of interest (e.g., age). The new subsection 5.2 elaborates on the use of non-central and central moments.

**Original comment:** L187. Need more explanation/description on "particle-based simulation".

**Answer:** A technical description of the model is added to the supplement section 1.1.

**Original comment:** L187-190. Difference between two experiments was attributed to step size in the Lagrangian simulation. This seems to be easy to check and I think the authors should provide some results on this. Also, if adding production/consumption terms are possible with the code as mentioned by the authors, why not checking the formulation based on those numerical experiments?

**Answer:** Properly evaluating the effect of changing the step size in the Lagrangian model is not so straightforward since D also depends on the step size. However, we have carried out additional analytical validation, now added to supplement section 2.

We have tested in simulations adding production and consumption reactions. In the code available online, interested readers can perform the same tests.

**Original comment:** L197-198. Eq. 30b and 30c may want to be explained in more detail with using more generalized equations (something similar to those in Section 3.1 but more generalized as done for diffusion)?

**Answer:** The derivations follow section 3, and we now explicitly refer to this section (application 1: line 228 "Refer to... these terms", application 2: lines 244-245 "The rate... these definitions", application 3: lines 279-280: "The age-dependent... specified as").

**Original comment:** L222. I thought you do not need lower boundary conditions assuming bioturbation is limited within top ~10 cm.

**Answer:** A model must always have a defined upper and lower boundary condition. However, the reviewer is correct that since the bioturbation will have dropped to 0, the zero gradient condition at the bottom (or another boundary condition) will practically not matter. For completeness, it is still good to mention the specified condition.

**Original comment:** L223. More details on 30-G model would be helpful.

**Answer:** We added a technical description of the model to the supplement section 1.2.

**Original comment:** L232-236. I think the explanation of production of moments is less organized in Section 3 than that for diffusion in Section 2. Could it be possible to make explanations/formulations easier to follow? This might include, e.g., giving more general formulation first and then presenting different cases later (like for diffusion, first Section 2 and then Table 1 and Appendix), giving the final forms of those production terms after the manipulation of equations, and so on.

**Answer:** The new section 3 in the manuscript should address this comment. The formulations containing binomial coefficients can be programmed into a script, making it unnecessary to write them out by hand.

**Original comment:** L244. As no values of w1-w3 are given, I have no clue what I am supposed to see in Fig. 4A.

**Answer:** The point was to compare the "real" distribution in black to a reconstruction based on a direct least-squares fit and an indirect fit based on the moments. In the revised manuscript we removed this figure entirely. Figure 3 has been completely changed and now shows the distributions that evolve at several depths in the model domain, which made Fig. 4 somewhat obsolete.

**Original comment:** L250. How are the weights determined? Assume they are determined rather arbitrarily after reading Discussion?

**Answer:** Considering the definition $\phi=(\chi-\mu)^q$, one can expect $\varepsilon_2$ to be of magnitude $\varepsilon_1^2$ and $\varepsilon_3$ to be of magnitude $\varepsilon_2^{3/2}$. $\varepsilon_2$ and $\varepsilon_3$ were thus scaled to similar magnitude. For $\varepsilon_1$, we divided through a smaller number to create a higher weight artificially.

The approach for the third application has changed substantially in the revised manuscript. We now use standard distributions (i.e., the translated Weibull and triangular distributions). An optimization scheme is no longer used. Now we use a root-solver without weighing the errors.

**Original comment:** L261. More details on "discrete simulations" would be helpful. Confusing because even continuum model does numerically integrate fitted function (L254-256) if I am not mistaken?

**Answer:** Yes, that is true. We have changed the word "discrete" at many locations to avoid confusion. Instead, we refer to validation simulation often as simulation based on bins. Also, the validation simulation was changed. The mismatch, depending on the number of bins, was caused by numerical diffusion. Please, refer to supplement section 1.3 for the new approach.

**Original comment:** L295. Does this difficulty come from numerical diffusion and/or grid size, i.e., numerical errors? Do numerical errors get bigger for higher order of moments?

**Answer:** For this example, diffusion and grid size do not play a role. In Figure 4 of the original manuscript, we directly fitted a hypothetical distribution (which was produced by interpolating a "discrete" distribution from an arbitrary simulation), treating it as the actual distribution. Hence, how this distribution was formed does not matter. The question is how well it can be reproduced based on the moments (mean, variance, and skewness). The point is that the extremes have much higher weight, as $(\chi-\mu)$ will be larger, which will scale exponentially with the order of moment due to the definition $\phi_q=(\chi-\mu)^q$. This is problematic since the extremes may not be that important for the overall reactivity. For instance, a tiny amount of ancient material will only slightly lower the overall

reactivity but greatly impact the higher moments. Therefore, simulating a higher number of moments may be unpractical.

We edited this part of the discussion, but the message did not change (lines 373-378: "A fundamental... higher-order moments.").

**Original comment:** L301. Or better/more stable solution seeking like with e.g., ensemble Kalman filter?

**Answer:** We have not tested this approach.

**Original comment:** Appendix A. More detailed derivations of Eqs. A10, A11, A12 would be desirable.

**Answer:** We added steps to these derivations and improved the explanation, as suggested.

**Original comment:** L18, 43. "Kuderer et al." should be replaced with "Kuderer".

**Answer:** We corrrected the mistake.

**Response to reviewer 2**

**Original comment:** I have read through the manuscript written by Rooze et al. This manuscript derived a new diffusion-advection-reaction type of partial differential equations based on centralized moment. Compared with Lagrangian frameworks, this Eulerian PDEs can be analytically evaluated and are computationally less expensive. The authors applied this PDEs to simulate organic matter age and reactivity with mixing processing in marine sediments. Overall, this work gives an opportunity to include bioturbation in the early diagenesis model with continuous distributions instead of the Multi-G model. However, the manuscript is not well organized and is difficult to follow, and it needs extensive revision to be accepted for publication.

**Answer:** We thank the reviewer for his suggestions and comments. We have changed the manuscript's organization by adding formal derivations of production/consumption terms in a new section (section 3 in the revised manuscript) and removed the corresponding derivations from the application section. We added a table for the variables, making the derivations easier to follow.

**Original comment:** In the introduction, the authors can emphasize the necessity of the continuous distributions with mixing processes, what is the current state of this model? Multi-G models have often been used to simulate the organic matter degradation with mixing processes including bioturbation. The challenges of such continuous model for bioturbation can be reviewed in the introduction.

**Answer:** Multi-G is a discrete approach often used to simulate mineralization in bioturbated sediments. The drawback of the multi-G approach is the somewhat arbitrary assignment of reactivity classes, which makes comparing different model parameterizations harder/impossible. We point this in the revision out in the introduction (lines 32-35: "The disadvantage... in reactivity."). At the same time, we do not want to overstate the point, as multi-G models still provide a simple and computationally efficient way to parameterize mineralization.

Eulerian models with continuous OM reactivity description have not accounted for the effect of bioturbation. Freitas et al. (2021) recently compiled parameters for the continuous gamma distribution in various study sites. However, for strongly bioturbated environments, they applied a discrete multi-G model. This inconsistency highlights a clear problem and poses several questions. As bioturbation typically dominates transport in the upper centimeters of sediment, applying a continuous model that cannot account for mixing may be mechanistically flawed, reducing its predictive power in most environments. In our test simulations, we find that bioturbation greatly impacts the distribution. We also added a sentence in the abstract to point this out.

**Original comment:** In whole section 2 and Appendix A, all equations were derived without any references except Crank (1956). I don't know which equations are newly derived and which are from literatures. Please check and explain them carefully.

**Answer:** When we provide derivations without citation, it implies we derived the equations ourselves. When we obtain an equation that other scientists obtained, we specifically mention that (see, for instance, end of sect. 2.2). For some very general equations, such as basic calculus, the equation for chemical (Fick's) diffusion, and the conversion of central to zero moments, we did not add references.

**Original comment:** In this manuscript, the authors have used confusing definition. For example, tau and mu are defined as particle age and averaged age (L68 - L69), but later tau=k is defined as reactivity (L192). k=f(tau) was also used (L227). tau was not even defined in int{tau^x C_tau dtau} (L172). Please make all definitions consistency.

**Answer:** We thank the reviewer for this suggestion. We have now changed $\tau$ and X everywhere to $\chi$. We also added a table (Table 1) in which the meaning of all variables is explained.

**Original comment:** I think the age defined in this manuscript is not age, rather transit time. The distribution function in age-structure model was usually expressed as f(tau,k,t) depending on decay reactivity k, age tau and time t. The average reactivity and age, and high-order moment can thus be calculated from f(tau,k,t). At the sediment-water interface, the age of organic matter will not be zero. The reactivity and age will have a distribution (e.g. gamma distribution) at the sediment-water interface. If this distribution is included in the model, could you derive the PDEs for the transit time, age, reactivity based on your model?

**Answer:** The word "age" is typically used in relation to organic matter reactivity. In physical transport models, "transit time" may be more common. However, there is not much difference conceptually, and it does not matter for the math or model implementation. We now mention "transit time" in the introduction (e.g., line 16: "For example... been simulated."), but generally, we stick to "age" for consistency.

We agree that the age will not be zero at the sediment-water interface, which is already reflected in our simulations.

The derived PDEs apply to any distribution. However, mixing and reaction processes will let the distribution evolve into a different shape, which needs to be reconstructed in order to perform the integrations for the reaction terms. The fundamental problem is that the change in distribution shape means it no longer conforms to the distribution type of the deposited matter. Thus, when a gamma distribution is used for the reactivity distribution of deposited OM, the distribution will no longer be a gamma distribution at depth due to mixing. The challenge is to find a distribution that may work well at different depths.

In this manuscript, we do not develop a method to reconstruct distributions resulting from deposited OM with the gamma distribution. However, we use different distributions, i.e., a newly invented distribution in the second application (eq. 35) and a triangular (eq. 38) and translated Weibull distribution (eq. 40) in the third application.

**Original comment:** Meile and Van Cappellen calculated particulate age and transit time distributions by particle tracking approach in bioturbated marine sediments. Could this model reproduce the results with same setup?

**Answer:** We have not directly compared our model results with this model. The Lagrangian simulation in the manuscript only serves to validate the diffusion equations (Table 1). The supplement provides additional proof and validation for the diffusion equations (supplement section 2). This, together with the simple Lagrangian model and mathematical derivations, gives sufficient evidence that the derivations are correct.

**Original comment:** The authors emphasized the important of center-moments to derive the PDEs. Please gives the difference in detail between center-moments and non-central moments and how they converted to each other.

**Answer:** We elaborate on using centralized and non-centralized moments in the equations for reactions (section 5.2, also see response to reviewer 1). A derivation for PDEs for the diffusion of non-central moments was already given by Delhez and Deleersnijder (2002), which we will not repeat. The conversion from central to non-central moments is given in equation 26 in the revised manuscript. However, the conversion of different moment types are well known and can be readily found on the internet (see, for instance, https://en.wikipedia.org/wiki/Central_moment). The revised manuscript distinguishes raw, non-central, and central moments and mentions the definitions in the text below equation 8. Raw and central moments are also defined in Table 1. In general, moments

can be described by $\langle(\chi-\psi)^q\rangle$, whereby raw, non-central, and central moments correspond to $\psi=0$, $\psi\neq\mu$, and $\psi=\mu$, respectively.

**Original comment:** In application 3.1 a particle tracking simulation has been used to compare current model, but the authors didn't mention how particle tracking simulation works in the text. Similar in application 3.2, it was not mentioned how 30-G model works. The same is in application 3.3 for the discrete simulation with 500 age bins.

**Answer:** We added the technical description of the validation models to the supplement (section 1). Please, note that the validation simulation for the third application fundamentally changed.

**Original comment:** I found one problem in current model is to choose a distribution function to fit the moments. This choice is not unique and it must be numerically evaluated and time-consuming. The multidimensional root-finding procedure may fail. I think that this model lacks generality to model early diagenesis at marine sediment. Maybe the authors can run example with the real data from marine sediment.

**Answer:** Indeed, a distribution may not be uniquely defined by a limited number of moments, as is discussed in lines 350-354 (before revision, lines 272-275): "In principle... always succeed". The largest practical drawback of the approach is that it may be hard to reconstruct a distribution from moments (for which we used the root-finding procedure, eq. 33). This problem is discussed in the text at length in section 5.1.

The simulations were stable and ran relatively fast for the application in section 4.2 (revised manuscript). Please, note that our primary goal was not to produce a fast model. Numerous optimization may be considered. However, the model has the advantage that it is easy to understand, and the published script is relatively short, which is better for communication purposes.

The third application was indeed very slow. However, the reviewer's comments encouraged us to improve this application. We now only use well-known distributions, i.e., the tridiagonal and translated Weibull distribution. The script has become more stable and runs well for these two distributions. Also, the implementation has significantly changed. We no longer use an ode-solver but switched to a classical PDE scheme. The root-solving procedure needs to be carried out less frequently in this new scheme. Now the simulations (for 50 cells and 50 years) runs with the tridiagonal and Weibull simulations in 10 s and 90 s, respectively, on a regular desktop machine (Intel(R) Core(TM) i7-6700 CPU @ 3.40GHz).

Most early diagenetic models use 1-G, 2-G, or 3-G formulations (Arndt et al., 2013). The second application demonstrates that our new model is up to the task, as it reproduces results also from a 30-G model. The third application also works well and could be used to simulate other reaction kinetics based on age.

**Original comment:** Although the author listed the application of the Eulerian model under three different conditions, the authors could select a specific site, including organic matter content data and organic matter 14C age data, to further validate the accuracy of the model.

**Answer:** This paper aims to derive general equations that can be used for reaction-transport models and are not limited to describing OM mineralization in sediments. An application of the model to measured organic contents, considering good dating proxies, and elaborating on the most suitable distribution for organic matter reactivity, can better be addressed in a separate paper.

**Original comment:** The mathematics is rather complex to understand. The meaning of many mathematical symbols is confusing (e.g., Line 59: What does $\delta x$ mean?). Therefore, I suggest that the author make a table showing in detail the meaning and value of each mathematical symbol in the text.

**Answer:** We have followed the reviewer advice. Please, see Table 1 in the revised manuscript.

**Original comment:** The setting and handling of the upper and lower boundaries of the model is the key to solving the model. However, the description of the upper and lower boundaries in the text is not very specific. I suggest listing more details and formulas to show how the upper and lower boundaries are handled in the solution.

**Answer:** It is a good point, and we have added text for each application to explain the boundary conditions. See, lines 229-230: "Here fixed... boundary conditions" for application 1, lines 264-266: "The uniform... boundary condition" for application 2, and lines 309-311: "The upper... the domain."

Sincerely,

Jurjen Rooze

PS: All cited references can be found in the first submitted manuscript.

---

## Author Response (AR2)

Dear reviewer,

We are again very grateful for the thorough review of our manuscript and the constructive comments, which contributed greatly to the quality of the revised manuscript. Below we respond to all the comments.

**Comment:** Revision by the authors in response to the previous review has been reasonably made in most cases and mathematical explanations and model confirmation and comparison are enhanced in my opinion. While I still found a couple of parts/equations difficult to understand, it is most likely that those can be easily addressed by the authors. I think after reflecting those the manuscript looks acceptable for GMD.

**Response:** We are glad that the reviewer appreciated our efforts to improve the manuscript and that he considers it to be acceptable for publication in GMD after minor revisions.

**Comment:** Eq. 25. Should right-hand side requires a factor of 1/V to be multiplied with?

**Response:** The reviewer is correct, and we made the change.

**Comment:** L175. Same as Eq. 25. The first definition of mu_x is understandable, but the second one with summation symbol is hard to understand especially given the definition in Table 1 and equations in L81 and 82. The second equation may require 1/V to be multiplied with on the right hand side? Eq. 26 remains valid regardless of above two points.

**Response:** Also, here, the reviewer is correct, and we made the change.

**Comment:** Eq. 28. Again to be consistent with previous definition of mu, C should not be here?

**Response:** This has now also been corrected. It was done correctly in the code and applications, so this mistake does not affect other parts of the manuscript.

**Comment:** L192. I cannot understand why r = 0 in Eq. 26 leads to R = P. Should this be like r = constant?

**Response:** For production, Eq. 26 is treated as an indefinite integral. When $r(\chi) = 0$ the result P is simply the constant of integration, which is correct for the zeroth-order kinetics considered. It would be incorrect to define r = constant, as this would yield a result in the form of $A\chi + B$ and, therefore, correspond to first-order kinetics.

We now clarify this in the text: "Representing the production of new material with a zeroth-order kinetic term (r = 0) results in R = P as the constant of integration when treating the integral in equation 27 as indefinite."

**Comment:** L201-202. "As these reactions do not discriminate with respect to age, the moments will not change." For this $r(x)$ has to be no longer a function of x as $r = -kC$; otherwise, $r(x) = -kC(x)$ and dphi/dt|R will be still a function of x and need distribution integration for its calculation? Regardless of whether this is correct or not, please clarify about parameterization of $r(x)$ too.

**Response:** If the reaction does not discriminate by age, then $f(\chi)$ is not a function of $\chi$ (more precisely, it is a constant function with $df/d\chi=0$). The reaction rate $r(\chi)$ still is a function of $\chi$: it is proportional to the concentration $C(\chi)$. Thus, the shape of the distribution $C(\chi)$ remains unchanged since the reaction $r(\chi)$ has the same shape. The magnitude can change (changing the total concentration C), but this does not affect the moments.

**Comment:** Appendix: Eq. A10. I still could not figure out how the first term on the right-hand side of Eq. A10 is derived from Eq. A9 and divergent theorem.

**Response:** The first term of equation A8 is

$$T = \frac{\partial \phi}{\partial \mu} \left( \frac{D_l C_L}{\delta_x} \frac{\partial \mu}{\partial x}\bigg|_l - \frac{D_r C_R}{\delta_x} \frac{\partial \mu}{\partial x}\bigg|_r \right)$$

The equations

$$C_L = C_l - \frac{1}{2}\frac{\partial C}{\partial x}\delta_x \qquad\qquad \text{(A9a)}$$

$$C_R = C_r + \frac{1}{2}\frac{\partial C}{\partial x}\delta_x \qquad\qquad \text{(A9b)}$$

are inserted, yielding

$$T = \frac{\partial \phi}{\partial \mu} \left( \frac{D_l C_l}{\delta_x} \frac{\partial \mu}{\partial x}\bigg|_l - \frac{D_r C_r}{\delta_x} \frac{\partial \mu}{\partial x}\bigg|_r \right) + \frac{\partial \phi}{\partial \mu} \left( -0.5 D_l \frac{\partial C}{\partial x}\frac{\partial \mu}{\partial x}\bigg|_l - 0.5 D_r \frac{\partial C}{\partial x}\frac{\partial \mu}{\partial x}\bigg|_r \right)$$

When we take the limit, for the first term, we obtain

$$\lim_{\delta x \to 0} \frac{\partial \phi}{\partial \mu} \left( \frac{D_l C_l}{\delta_x} \frac{\partial \mu}{\partial x}\bigg|_l - \frac{D_r C_r}{\delta_x} \frac{\partial \mu}{\partial x}\bigg|_r \right) = -\frac{\partial \phi}{\partial \mu} \left[ \frac{\partial}{\partial x} \left( DC\frac{\partial \mu}{\partial x} \right) \right]$$

For infinitesimally small x, the terms inside the brackets of the second term may be added, yielding

$$\frac{\partial \phi}{\partial \mu} \left( -0.5 D_l \frac{\partial C}{\partial x}\frac{\partial \mu}{\partial x}\bigg|_l - 0.5 D_r \frac{\partial C}{\partial x}\frac{\partial \mu}{\partial x}\bigg|_r \right) = -D\frac{\partial \phi}{\partial \mu}\frac{\partial C}{\partial x}\frac{\partial \mu}{\partial x}$$

Hence,

$$\lim_{\delta x \to 0} T = -\frac{\partial \phi}{\partial \mu} \left[ \frac{\partial}{\partial x} \left( DC\frac{\partial \mu}{\partial x} \right) + D\frac{\partial C}{\partial x}\frac{\partial \mu}{\partial x} \right]$$

which is the first term of A10.

Since these steps do not introduce new ideas, we do not want to write them out in the manuscript. However, the description may not have been entirely accurate, as we do not have to apply the divergence theorem, but instead, we take the limit to 0. We have improved the text in the revised manuscript.

**Comment:** Supplement: 1.1. It is very hard to imagine what kinds of vectors the authors are using. Definition of x is given but xL and xR are assumed to be obvious and not defined. Individual x vector contains ages of individual n particles? If so and x0 is x at time 0, it is confusing as xL and xR appear in definition of x0.

**Response:** The text's description of $\chi_L$ and $\chi_R$ was not sufficiently clear. These vectors contain the ages of all particles at the domain's boundaries. It is correct that $\chi_0$ is $\chi$ at time 0. We revised the text in the supplement and now show the equations for the boundary conditions (equations S2 and S3).